🍥 PLOS | ONE

# Ovine prenatal growth-restriction and sex influence fetal adipose tissue phenotype and impact postnatal lipid metabolism and adiposity in vivo from birth until adulthood

Jacqueline M. Wallace ⬤ *, John S. Milne, Beth W. Aitken, Raymond P. Aitken, Clare L. Adam

Rowett Institute, University of Aberdeen, Aberdeen, United Kingdom

* Jacqueline.Wallace@abdn.ac.uk

**Data Availability Statement:** All relevant data are within the paper and it Supporting Information files.

## Abstract

Adipose tissue development begins *in utero* and is a key target of developmental programming. Here the influence of nutritionally-mediated prenatal growth-restriction on perirenal adipose tissue (PAT) gene expression and adipocyte phenotype in late fetal life was investigated in both sexes in an ovine model. Likewise circulating leptin concentrations and non-esterified fatty acid (NEFA) and glycerol responses to glucose challenge were determined in relation to offspring adiposity at key stages from birth to mid-adult life. In both studies' singleton-bearing adolescent sheep were fed control or high nutrient intakes to induce normal or growth-restricted pregnancies, respectively. Fetal growth-restriction at day 130 of gestation (32% lighter) was characterised by greater body-weight-specific PAT mass and higher PAT expression of peroxisome proliferator-activated receptor gamma (*PPARγ*), glycerol-3-phosphate dehydrogenase, hormone sensitive lipase (*HSL*), insulin-like growth factor 1 receptor, and uncoupling protein 1. Independent of prenatal growth, females had a greater body-weight-specific PAT mass, more multilocular adipocytes, higher leptin and lower insulin-like growth factor 1 mRNA than males. Growth-restricted offspring of both sexes (42% lighter at birth) were characterised by higher plasma NEFA concentrations across the life-course (post-fasting and after glucose challenge at 7, 32, 60, 85 and 106 weeks of age) consistent with reduced adipose tissue insulin sensitivity. Circulating plasma leptin correlated with body fat percentage (females>males) and restricted compared with normal females had more body fat and increased abundance of *PPARγ;, HSL*, leptin and adiponectin mRNA in PAT at necropsy (109 weeks). Therefore, prenatal nutrient supply and sex both influence adipose tissue development with consequences for lipid metabolism and body composition persisting throughout the life-course.

## Introduction

Appropriate prenatal adipose tissue development to facilitate non-shivering thermogenesis at birth is fundamental to the immediate survival of precocial mammals [1,2]. Thereafter adipose

**Funding:** This work was funded by the Scottish Government's Rural and Environmental Science and Analytical Services Division (RESAS) including the Strategic Partnership for Animal Science. The funders had no role in study design, data collection and analysis, decision to publish, or preparation of the manuscript.

**Competing interests:** The authors have declared that no competing interests exist.

tissue is central to energy metabolism and developmentally programmed alterations in adipose tissue physiology are considered a common phenotypic feature underlying a range of issues that become apparent in postnatal life [3]. In humans these include metabolic, cardiovascular and reproductive disorders [4–6], while in meat producing species excess body fat negatively impacts meat quality and hence financial returns to the producer [7,8].

In humans and sheep, fat is mainly deposited around the fetal kidneys from mid-pregnancy onwards, and by late gestation this perirenal adipose tissue (PAT) depot contains both unilocular (white-lipid storage) and multilocular (brown-thermogenic) adipocytes [9–11]. The latter cells are mitochondria-rich and high levels of uncoupling protein 1 (UCP1) burn fatty acids and glucose to release heat in the newborn [12]. Immediately after birth the PAT depot of lambs is transiently depleted of lipid and as the neonate suckles and grows the brown adipocytes are lost and white lipid-filled adipocytes dominate [11]. In adult life, humans and sheep are similar with respect to mature body weight and adiposity.

Maternal dietary intake is a key driver of fetal nutrient supply and as such is likely to play an important role in establishing the size of the prenatal PAT depot and its cellular make-up, with potential consequences for metabolism and body composition across the life-course. In spite of this assertion sheep models involving either under- or over-nourishing adult dams during defined periods of pregnancy have failed to reach a consensus with regard to the impact of altered nutrient supply on fetal growth, PAT depot mass or expression of genes involved in adipose tissue growth, differentiation or function when measured in late gestation [13–20]. This may reflect inconsistencies both between and within studies with respect to maternal parity, dietary history, numbers of rams used, fetal number, sex distribution and season [2]. Most of these factors are kept constant in our adolescent sheep models: these involve assisted conception procedures using one sire and transfer of single high-quality embryos harvested from donors with a known nutritional and reproductive background into young primiparous adolescent recipients of equivalent age, weight and adiposity at conception. Thereafter the adolescents are nutritionally manipulated throughout gestation with the control or reference group being mothers whose initial adiposity level is maintained all the way through. In the *undernourished model*, further maternal growth during pregnancy is prevented and the gradual depletion of her body reserves limits fetal nutrient supply, independent of any change in placental size [21]. By late gestation the fetuses are modestly lighter with lower PAT mass and fewer unilocular adipocytes [22].

Our focus here is our *overnourished model* where a high dietary intake throughout gestation promotes continued maternal growth and increased adiposity but where placental growth, uteroplacental blood flows and fetal nutrient delivery are attenuated leading to a high incidence of marked fetal growth-restriction by late gestation [reviewed in 23]. These effects have proved robust across multiple studies with ~50% of offspring of overnourished mothers having a birthweight >2SD below the mean birthweight of the normally-growing reference group. Studying the relative development of the fetal adipose tissue in these pregnancies is particularly pertinent as the lambs are born prematurely (~5 days early), and initial colostrum supply is severely impaired, making them vulnerable to hypothermia and starvation. We hypothesised that this form of nutritionally-induced placental insufficiency and poor fetal nutrient supply would impact the size and adipocyte composition of the fetal PAT depot and this would be associated with changes in the expression of selected molecular markers of adipocyte proliferation, differentiation and function (Experiment 1). We further postulated that the adipose tissue phenotype of prenatally growth-restricted lambs would influence their postnatal lipid metabolism and body composition, and accordingly we measured circulating leptin concentrations and non-esterified fatty acid (NEFA) and glycerol responses to glucose challenge in relation to offspring adiposity at key stages from birth to mid-adult life (Experiment 2). Given the

pronounced sexually dimorphic responses reported postnatally in the ovine nutritional programming literature [24–30], both studies were sufficiently large to allow the impact of fetal or offspring sex to be examined.

## Materials and methods

### Establishment of pregnancy and nutritional management

All procedures were licensed under the UK Animals (Scientific Procedures) Act of 1986 and approved by the Rowett Institute's Ethical Review Committee. Ewes were housed in individual open-wide bar pens that facilitated nose to nose contact with neighbouring animals, under natural lighting conditions at 57˚N, 2˚W. Singleton pregnancies were generated following super-ovulation and laparoscopic intrauterine insemination (single Dorset Horn sire) of adult donor ewes (third parity, Border Leicester x Scottish Blackface, mean adiposity score 2.3 units), and synchronous transfer of the resulting embryos into adolescent recipients (~8.5 months old, Dorset Horn x Greyface) of similar initial weight and adiposity. For recipients, oestrus synchronisation commenced on Day -15 relative to the onset of oestrus (Day 0) by insertion of vaginal sponges containing 60mg medroxyprogesterone acetate (Veramix®, Intervet UK Ltd, Welwyn Garden City, Herts, UK). Thirteen days later (Day -2) at 14:00h, sponges were removed, and ovulation stimulated by i.m. administration of 600 iu pregnant mare serum gonadotrophin (PMSG, Intervet UK Ltd). For donor ewes, synchronisation commenced on Day -14, when vaginal sponges containing 40mg fluorogestone acetate (Chronogest®, Intervet UK Ltd) were inserted. Seven days later (Day -7) these sponges were removed and immediately replaced with fresh ones of the same type. A superovulation regime was commenced on Day -4 at 08:00h with the i.m. administration of 1.125mg ovine follicle-stimulating hormone (oFSH, Ovagen®, Immuno-Chemical Products Ltd, Auckland, New Zealand) immediately followed by 125µg IM cloprostenol (Estrumate®, Schering-Plough Ltd, Welwyn Garden City, Herts, UK), a synthetic prostaglandin-F2α. At 18:00h a second dose of oFSH was given together with 400iu PMSG. Further twice daily doses of oFSH were administered at 08:00h and 18:00h on the following three days (Day -3 to -1) and sponges were removed on Day -2 at 18:00h. The onset of oestrus was assessed by presenting females to vasectomised rams three times daily to ensure optimum synchrony between potential donor and recipient animals. On Day 0 at 16:00h, donor ewes underwent insemination directly into the uterine cavity under direct laparoscopic visualisation, as previously described [31]. In both studies this involved fresh semen, collected by artificial vagina from the same sire of proven fertility, and diluted 1:3 with phosphate-buffered saline. Approximately 0.3ml of diluted semen was deposited in each horn. Four days following insemination, multiple embryos were recovered from donor ewes at laparotomy using a standard surgical technique of retrograde flushing of each oviduct. This was achieved by advancing a fine glass cannula into the fimbria, injecting ovum culture media (ICN Biomedicals, Ohio, USA) warmed to 37˚C into the lumen of the ipsilateral uterine horn ~ 5cm from the utero-tubal junction and milking the media out through the cannula into a sterile glass embryo dish. Developmental stage and quality were assessed using a stereomicroscope and the embryos held at 33˚C in fresh ovum culture media until transferred to recipients within 4h of recovery. Embryos of optimum quality and stage were synchronously transferred in singleton into the uteri of adolescent recipients using a laparoscopically-assisted technique. Ovulation rate was determined, and the embryo transferred into the tip of the uterine horn ipsilateral to the ovary with the greater number of corpora lutea.

In Experiment 1 (fetal study), the adolescent recipients had a mean liveweight of 44.3 ±0.34kg, and an adiposity score of 2.3±0.01 (equivalent to 23% body fat, based on a 5-point scale, where 1 = emaciated and 5 = extremely obese; [32]) at the time of embryo transfer. In

Experiment 2 (postnatal study) recipient liveweight and adiposity score were 46.6±1.09kg and 2.3±0.04 units, respectively.

In both experiments, and commencing directly after embryo transfer, adolescent recipients were offered either a control or high level of a complete diet providing 12 MJ of metabolisable energy (ME) and 140 g of crude protein per kg. Diet composition was 30% coarsely milled hay, 41.5% barley 17.5% Hipro soya, 10% molasses, 0.35% salt, 0.25% limestone, 0.25% dicalcium phosphate and 0.15% of a vitamin-mineral mix [33]. Fresh food was offered in two daily feeds at 08:00h and 16:00h. For the control group the dietary level was calculated to preserve the original maternal adiposity level throughout pregnancy and to provide 100% of the estimated metabolisable energy and protein requirements of the adolescent sheep carrying a singleton fetus according to stage of pregnancy ([34]; normal fetoplacental growth). In contrast, the high ration which was fed *ad libitum*, was intended to support continued maternal growth and increasing adiposity at the expense of the conceptus: to achieve this, recipients had the ration increased stepwise over a 2-week period until the daily food refusal was ~15% of the amount offered. These animals were considered overnourished (~2.25 x control). To facilitate accurate nutritional management external adiposity scores and weights were measured fortnightly throughout pregnancy.

## Experimental design

In Experiment 1 viable pregnancies were established in 28 control and 32 overnourished dams (measured by transabdominal ultrasound at d45 and d90). A maternal blood sample was collected just before necropsy on day 130 of gestation and the plasma used to confirm metabolic status (glucose, insulin and NEFA concentrations). Ewes were killed by i.v. sodium pentobarbitone (20ml Euthesate; 200mg pentobarbitone/ml; Willows Francis Veterinary, Crawley, UK) and exsanguination. The gravid uterus was weighed, opened, and fetal blood sampled by cardiac puncture immediately before administering intracardiac sodium pentobarbitone (3ml Euthesate); this plasma was analysed for insulin. The fetus was dried and weighed, and the PAT rapidly dissected free from the kidneys. PAT was weighed and samples either snap-frozen in isopentane chilled by liquid nitrogen and stored at -80˚C pending gene expression analysis (n = 12 genes) or fixed in 10% neutral buffered formalin and embedded in paraffin for histological quantification of adipocytes. Intact placentomes were dissected and weighed as was the fetal brain, fetal liver and maternal carcass: these weights helped confirm the efficacy of the nutritional treatments.

A full description of the derivation of the postnatal animals studied in the course of Experiment 2 has been reported elsewhere with respect to pregnancy outcome, neonatal care and postnatal management [35]. Briefly 24 normal birthweight lambs from control-fed ewes (12 females and 12 males) and 25 growth-restricted lambs from overnourished ewes (16 females and 9 males) were studied from birth forwards. Males remained gonad intact. During lactation the mothers were fed the complete diet described above *ad libitum*. Offspring (lambs) were weaned at 11 weeks of age and had free unlimited access to the complete diet throughout postnatal life. Offspring underwent intravenous glucose tolerance tests (GTT) at 7, 32, 60, 85 and 106 weeks of age as previously described for suckling [29] and adult life-stages [36]. At the 7-week stage this involved a 3 hour fast prior to i.v. glucose being administered at 0.25g/kg body weight. At subsequent ages offspring were fasted overnight for 18.5 hours prior to i.v. glucose at 0.5g/kg body weight. Blood samples were collected at -30, -15, 0, +5, +10, +15, +30, +45, +60, 90 and +120 minutes at 7 weeks of age and a further sample was collected at +180 minutes at all other stages. Insulin and glucose responses have already been reported and herein NEFA and glycerol were measured. Fasting levels (baseline), slope (from 10 to 30 minutes), and areas

under the response curve (AUC) were determined. AUC was calculated as integrated plasma concentrations following glucose administration [5–120 or 5–180 minutes overall, for 7 weeks versus all other stages, respectively] above the mean baseline [-30 to 0 minutes] concentrations. In addition, venous blood samples collected at ~11:00h were collected at weekly then fortnightly intervals throughout the life-course and leptin concentrations measured in the resulting plasma. Plasma leptin and adiponectin were also measured in a blood sample collected immediately before necropsy. Body composition was determined by dual energy X-ray absorptiometry (DEXA) under general anaesthesia at 11, 41, 64 and 107 weeks of age as previously described [35]. The body fat data are reproduced here to facilitate comparison with peripheral leptin concentrations, and both the body fat and bone mineral density data were used in an analysis (detailed below) to estimate body composition at birth and the average change thereafter. Off-spring were killed at 109–110 weeks of age as specified above and the major organs dissected. The PAT was weighed, sampled as for the fetal study and the expression of 10 genes determined.

## Fetal adipocyte histology

The procedure used to quantify the density of unilocular and multilocular cells in fetal PAT collected in Experiment 1 was similar to that reported previously [37]. Sections were cut (5μm), dried overnight, stained with haematoxylin and eosin, and viewed at 200x magnification using a Leica microscope. Ten separate complete fields of view per animal, ~1mm apart on a single section, were captured by digital camera and the images analysed using Image-Pro Plus (version 4.5.1, Media Cybernetics, Inc., Silver Spring, MD, USA). Standard point counting techniques [38] were employed: a standard grid was used to determine the adipose tissue component (i.e. unilocular or multilocular cell) falling below each of 45 grid points per image, thus totalling 450 points per animal. The volume density ($V_d$) of each cell type was calculated as $V_d = N/T$, where N is the number of points falling on unilocular or multilocular cells, and T is the total number of points counted. The total mass of the unilocular or multilocular component was calculated by multiplying the $V_d$ of each component by the PAT mass. Relative unilocular (or multilocular) fat mass (g/kg fetus) was determined by dividing by fetal weight. For each image the circumference of the largest unilocular cell per field of view was measured by manual tracing. This was used to generate an overall mean for the 10 fields of view examined per animal. The above adipose tissue cell types and circumference measurements were also quantified on a quality control slide on 10 occasions (~ between every 6th animal) and the resulting coefficients of variation for the 3 parameters ranged from 2–3%.

## Plasma analysis

Plasma insulin and leptin concentrations were measured in duplicate by radioimmunoassay [39,40]. The sensitivities of the assays were 0.17ng insulin/ml and 0.1ng leptin/ml and inter- and intra-assay coefficients of variation where applicable were less than 10%. Adiponectin concentrations were measured in duplicate using a sheep specific Elisa accordingly to the manufacturer's instructions (Catalogue number CSB-EL001366SH, www.cusabio.co). Samples were diluted 200 x, sensitivity was 0.6μg adiponectin/ml and duplicate variation was <10%. Plasma NEFA and glycerol concentrations were measured using an automated clinical analyzer with kits supplied by the manufacturer (Labmedics, Manchester, UK, duplicate variation <10%), and glucose by dual-biochemistry analyser (YSI model 2700, Yellow Springs, OH, USA; duplicate variation <3%).

## Quantitative real-time reverse transcription-polymerase chain reaction analysis

Messenger RNA for genes variously involved in adipocyte proliferation, differentiation and function were measured by quantitative real-time reverse transcription-polymerase chain reaction (qRT-PCR). Thus *PPARγ*, glycerol-3-phosphate dehydrogenase *(G3PDH)*, lipoprotein lipase *(LPL)*, fatty acid synthase *(FASN)*, *HSL*, leptin, adiponectin, *IGF1* and *IGF2* were measured in PAT from both fetal and postnatal studies; *UCP1*, *IGF1-R* and *IGF2-R* were measured in fetal tissue only and *INS-R* was measured in postnatal tissue only. Probe and primer sets for sheep-specific sequences of these genes were predominately as previously described [30,41]. The sequences of the cDNA forward and reverse primers and the Taqman probes for genes not reported previously were as follows: adiponectin, `5'- ACGGCACCACTGGCAAA-3'` (accession number KJ159213), `5'- TAGACGGTAATGTGGTAGGAGAAGTAGT-3'` and `5'(6FAM)- TCCTCTGCAATATCCCCGGGCTG-(TAMRA)3'`, respectively; *INS-R*, `5'- ACCGCCAAGGGCAAGAC -3'` (accession number AJ844652.1), `5'- AGCACCGCTCCACAAACTG -3'` and `5'(6FAM)- AACTGCCCTGCCACTGTCATCAACG -(TAMRA)3'`, respectively. In brief total RNA was extracted from 100mg frozen PAT using RNeasy Lipid Tissue Mini Kit (Qiagen, Crawley, West Sussex, UK). The quality and quantity of total RNA were determined via capillary electrophoresis using an Agilent 2100 Bioanalyzer (Agilent Technologies, Wilmington, DE, USA). Real-time RT-PCR reagents, probes, and primers were purchased from and used as recommended by Applied Biosystems (Warrington, UK). For each sample 54ng total RNA was subjected to reverse transcription (RT) in triplicate to generate first-strand cDNA using Taqman Reverse Transcription Reagents and Multiscribe Reverse Transcriptase. Polymerisation and amplification reactions for each RT sample were performed in duplicate in 20μl final volume using the Applied Biosystems 7500 Fast Real-Time PCR system. Quantification was performed by means of a relative standard curve method with serial dilutions of reference standard cDNA produced from RNA pooled from PAT of control and overnourished fetuses (Experiment 1, n = 3 per nutritional group and sex) or from control and prenatally growth-restricted adult offspring (Experiment 2, n = 3 per birthweight category and sex). Individual mRNA levels of genes of interest were expressed relative to the sample's own internal 18S RNA, determined using human 18S Pre-developed TaqMan Assay Reagents. Within studies samples were randomised to ensure that each nutritional treatment/ growth category and sex was represented in each 96-well plate. A quality control sample generated from the study-specific RNA pool was run on each plate and used to calculate inter- and intra-assay coefficients of variation (cov). Intra-plate cov varied for individual genes varied from 3.6 to 8.1% (overall mean±sem, 5.7±0.42%) and inter- plate cov varied from 0.9 to 9.8% (overall mean±sem 4.9±0.85%).

## Power calculation

The power calculation for the fetal study was based on the prediction that maternal overnutrition would impact fetal weight specific perirenal fat deposition by late gestation. This indicated that 25 animals per group would be required for the study to have 90% power to detect (at 5% significance) a 20% change in perirenal fat mass of 0.98 g/kg fetus assuming animal variability of 1.04 g/kg fetus. This was based on fetal perirenal fat mass data from a mixed population of female and male fetuses from control-fed pregnancies of equivalent genotype to that used here and at the same stage of gestation. The power calculation did not consider the sex of the fetus which was out-with our control but instead assumed that the sex distribution would be broadly similar within the nutritional treatment groups and thereby allow fetal sex to be included as a factor in the subsequent analysis. The power calculation for the postnatal study was based on

**Table 1. Maternal weight and adiposity change, and metabolic status at day 130 of gestation in relation to maternal nutrition and prenatal growth category (Experiment 1).**

| Gestational intake | Control | Overnourished | | P value[ß] |
|---|---|---|---|---|
| Growth status | Normal | Non-Restricted | Restricted | |
| Number of pregnancies | 28 | 16 | 16 | |
| Delta maternal weight[¥], Kg | 9.9±0.51[a] | 29.8±0.78[b] | 30.1±1.08[b] | <0.001 |
| Delta maternal BCS[♭], units | 0±0.0[a] | 0.6±0.03[b] | 0.6±0.03[b] | <0.001 |
| Maternal carcass weight, Kg | 36.3±1.02[a] | 51.3±0.86[b] | 50.9±0.75[b] | <0.001 |
| Maternal insulin, ng/ml | 0.9±0.08[a] | 2.1±0.20[b] | 2.1±0.17[b] | <0.001 |
| Maternal glucose, mmol/l | 3.1±0.09[a] | 3.6±0.09[b] | 3.5±0.12[ab] | 0.001 |
| Maternal NEFA, mmol/l | 0.300±0.0360[a] | 0.193±0.0299[ab] | 0.106±0.0119[b] | <0.001 |

[ß]*Post hoc* comparisons (Tukeys method) were used to further differentiate between the three groups, thus within rows values with different superscript letters differ at P<0.01.

[¥]difference between weight at necropsy minus the gravid uterus, and weight at embryo transfer;

[♭]change in external body condition score (BCS) from embryo transfer to necropsy.

previous studies that demonstrate that overnourishing pregnant adolescents negatively impacts lamb birthweight independent of sex. We assumed a similar number of females and males would be produced within each nutritional treatment group and thus 11 animals of each sex per group were selected in order for the study to have 90% power to detect a 20% decrease in birthweight of 1085g, assuming animal variability of 748g. This was based on lamb birth-weight data from control-fed pregnancies of equivalent genotype and summarized for several studies [23]. Embryo transfer is used to derive the viable pregnancies in this animal model and although we estimate the average conception rate to embryo transfer as ~70% and build in a loss-to-study at this initial stage, the overall conception rate can vary between studies, and accordingly, it is our practice to study all available animals irrespective of the power calculation.

## Data analysis

Data are presented as means ± standard error of the mean (SEM) and all statistical comparisons were made using Minitab (version 19; Minitab Inc., State College, PA). The individual animal data underlying these means are available (S1 and S2 Tables). All data were checked for normality using an Anderson-Darling test. Where the P value was <0.05, this was due to positively skewed distributions and so the data were log transformed before analysis. In Experiment 1 the fetuses of overnourished dams were classified as growth-restricted when fetal weight at necropsy was >2SD below the genetically-matched control group mean; the cut-off was 3676g (n = 16). The remaining overnourished fetuses were considered non-restricted (n = 16). The three groups (control [n = 28], non-restricted and restricted) were compared using one-way ANOVA for the maternal data (Table 1) and for selected fetal data when females and males were considered together as presented in the text. The fetal data presented in Tables 2 and 3 were compared by a general linear model (GLM) ANOVA as this approach allowed the effects of maternal gestational intake/growth status and sex, and their potential interaction to be compared. *Post hoc* comparisons (Tukey's method) were used to further differentiate between groups in both cases. Within individual groups paired Student's t tests were used to determine differences in fetal adipocyte cell type.

In Experiment 2 offspring variables measured at several ages throughout the life-course were primarily analyzed using a mixed-effects repeated-measures model with offspring number

**Table 2. Conceptus phenotype and adipose cell distribution in the fetal perirenal fat depot at 130 days of gestation in relation to maternal nutrition, prenatal growth category and sex (Experiment 1).** Values are group means ±sem.

| Gestational intake | Control | | Overnourished | | | | | | |
|---|---|---|---|---|---|---|---|---|---|
| Growth status | Normal | | Non-Restricted | | Restricted | | P value[¥] | | |
| Fetal sex | Female | Male | Female | Male | Female | Male | GI/growth | Sex | Interaction |
| Number | 19 | 9 | 8 | 8 | 8 | 8 | | | |
| Gestational age, d | 130.5±0.33 | 131.0±0.62 | 131.1±0.44 | 130.1±0.44 | 130.8±0.39 | 131.5±0.50 | 0.536 | 0.903 | 0.228 |
| Fetal wt, g | 4470±85$^b$ | 5192±142$^a$ | 4673±221$^{ab}$ | 4701±226$^{ab}$ | 2955±137$^c$ | 3216±210$^c$ | **<0.001** | **0.015** | 0.082 |
| Placentome wt, g | 490±19$^{ab}$ | 526±23$^a$ | 411±27$^{ab}$ | 434±34$^b$ | 229±17$^c$ | 270±25$^c$ | **<0.001** | 0.116 | 0.949 |
| Fetal:placentome wt | 9.27±0.26$^c$ | 10.00±0.48$^{bc}$ | 11.58±0.55$^{ab}$ | 11.12±0.71$^{abc}$ | 13.13±0.58$^a$ | 12.29±0.96$^{ab}$ | **<0.001** | 0.681 | 0.308 |
| Brain:liver wt | 0.303±0.012$^c$ | 0.285±0.013$^c$ | 0.291±0.017$^c$ | 0.336±0.021$^{bc}$ | 0.461±0.024$^a$ | 0.390±0.037$^{ab}$ | **<0.001** | 0.395 | **0.042** |
| Fetal insulin, ng/ml | 0.54±0.041 | 0.53±0.046 | 0.52±0.045 | 0.36±0.028 | 0.40±0.051 | 0.36±0.057 | **0.007** | 0.081 | 0.297 |
| Perirenal fat wt, g | 24.7±0.86$^a$ | 22.04±0.95$^{ab}$ | 21.33±1.21$^{ab}$ | 21.93±1.75$^{ab}$ | 17.01±1.32$^b$ | 16.88±1.30$^b$ | **<0.001** | 0.474 | 0.346 |
| Perirenal fat, g/kg fetus | 5.56±0.22$^a$ | 4.28±0.25$^b$ | 4.56±0.169$^{ab}$ | 4.66±0.30$^{ab}$ | 5.82±0.46$^a$ | 5.29±0.28$^{ab}$ | **0.016** | **0.023** | 0.065 |
| Unilocular fat | | | | | | | | | |
| Volume density, % | 40.6±1.45$^{b*}$ | 41.0±1.97$^{b*}$ | 43.7±2.50$^{ab*}$ | 50.9±2.54$^a$ | 42.3±1.79$^{ab*}$ | 47.7±2.18$^{ab}$ | **0.006** | **0.014** | 0.207 |
| Total mass, g | 10.0±0.50$^{ab¥}$ | 9.1±0.73$^{ab¥;}$ | 9.3±0.78$^{ab¥;}$ | 11.3±1.27$^a$ | 7.2±0.64$^{b¥;}$ | 8.1±0.78$^{ab}$ | **0.007** | 0.311 | 0.151 |
| Relative mass, g/kg fetus | 2.26±0.12$^ß$ | 1.78±0.17$^ß$ | 1.99±0.12$^ß$ | 2.40±0.23 | 2.47±0.22$^ß$ | 2.52±0.17 | **0.029** | 0.970 | **0.038** |
| Largest unilocular cell/image, pixcels | 517±9.2 | 535±19.8 | 519±16.5 | 540±16.8 | 519±16.8 | 535±13.1 | 0.977 | 0.145 | 0.990 |
| Multilocular fat | | | | | | | | | |
| Volume density, % | 59.4 ±1.45$^{a*}$ | 59.0±1.97$^{a*}$ | 56.3±2.50$^{ab*}$ | 49.0±2.54$^b$ | 57.7±1.79$^{ab*}$ | 52.3±2.18$^{ab}$ | **0.006** | **0.014** | 0.204 |
| Total mass, g | 14.7±0.63$^{a¥;}$ | 12.9±0.48$^{ab¥;}$ | 12.0±0.83$^{abc¥;}$ | 10.6±0.78$^{bc}$ | 9.8±0.78$^{bc¥;}$ | 8.8±0.71$^c$ | **<0.001** | **0.030** | 0.889 |
| Relative mass, g/kg fetus | 3.3±0.15$^{aß}$ | 2.5±0.11$^{cß}$ | 2.6±0.16$^{bcß}$ | 2.3±0.15$^c$ | 3.3±0.28$^{abß}$ | 2.8±0.19$^{abc}$ | **0.007** | **0.001** | 0.423 |

[¥]*Post hoc* comparisons (Tukey's method) were used to further differentiate between the six groups, thus within rows values with different superscript letters differ at P<0.05. Within individual groups adipose cell parameters with the same symbol (*,¥;, ß) differ from each other, P<0.05 to <0.001. Significant P values are highlighted in bold. Gestational intake, GI

**Table 3. Fetal perirenal fat gene expression at 130 days of gestation in relation to maternal nutrition, prenatal growth category and sex (Experiment 1).**

| Gestational intake | Control | | Overnourished | | | | | | |
|---|---|---|---|---|---|---|---|---|---|
| Growth status | Normal | | Non-Restricted | | Restricted | | P value[¥] | | |
| Fetal sex | Female | Male | Female | Male | Female | Male | GI/growth | Sex | Interaction |
| Number | 19 | 9 | 8 | 8 | 8 | 8 | | | |
| 18s | 0.022±0.001 | 0.019±0.001 | 0.022±0.001 | 0.022±0.001 | 0.023±0.001 | 0.022±0.001 | 0.102 | 0.428 | 0.482 |
| PPARɣ:18s | 11.68±0.594$^b$ | 12.83±0.972$^{ab}$ | 11.49±0.877$^{ab}$ | 12.36±0.938$^{ab}$ | 14.21±1.45$^{ab}$ | 15.58±0.81$^a$ | **0.007** | 0.152 | 0.970 |
| G3PDH:18s | 26.8±1.32$^c$ | 29.31±2.71$^{bc}$ | 30.73±1.53$^{abc}$ | 26.34±1.90$^c$ | 39.42±2.64$^a$ | 36.83±1.83$^{ab}$ | **<0.001** | 0.377 | 0.185 |
| LPL:18s | 24.11±1.69 | 25.38±3.99 | 23.26±2.62 | 20.10±1.86 | 20.30±2.27 | 26.01±4.92 | 0.566 | 0.603 | 0.383 |
| FASN:18s | 18.41±1.50 | 21.45±3.92 | 16.81±2.38 | 19.80±2.77 | 12.65±2.08 | 21.25±5.93 | 0.381 | 0.155 | 0.572 |
| HSL:18s | 22.43±1.41$^b$ | 27.13±2.69$^{ab}$ | 29.54±3.29$^{ab}$ | 28.03±2.06$^{ab}$ | 31.87±3.18$^a$ | 34.98±2.73$^a$ | **0.003** | 0.221 | 0.448 |
| Leptin:18s | 24.84±1.64 | 20.42±1.19 | 27.04±2.03 | 21.20±2.04 | 27.47±2.66 | 24.11±2.02 | 0.297 | **0.010** | 0.854 |
| Adiponectin:18s | 19.39±1.17 | 18.54±2.19 | 19.87±1.94 | 16.90±1.87 | 18.64±2.04 | 20.70±2.20 | 0.812 | 0.705 | 0.455 |
| IGF1:18s | 20.65±2.27$^a$ | 22.63±2.89$^a$ | 16.74±0.95$^{ab}$ | 21.29±2.58$^a$ | 12.61±1.51$^b$ | 19.52±2.74$^{ab}$ | **0.034** | **0.009** | 0.348 |
| IGF2:18s | 21.82±1.36 | 24.46±3.57 | 20.81±2.04 | 22.43±1.10 | 18.41±1.83 | 22.76±3.04 | 0.527 | 0.127 | 0.826 |
| IGF1R:18s | 30.05±2.84 | 30.01±2.47 | 32.00±3.07 | 32.68±3.97 | 36.75±3.15 | 37.41±3.04 | **0.019** | 0.624. | 0.903 |
| IGF2R:18s | 28.67±1.59 | 28.95±1.97 | 30.71±2.57 | 32.72±3.23 | 32.25±2.57 | 34.75±3.02 | 0.157 | 0.458 | 0.945 |
| UCP1:18s | 31.9±3.63$^b$ | 35.55±4.04$^b$ | 45.21±5.07$^b$ | 34.59±6.53$^b$ | 77.31±6.77$^a$ | 49.02±4.30$^b$ | **<0.001** | **0.007** | **0.009** |

[¥]*Post hoc* comparisons (Tukeys method) were used to further differentiate between the six groups, thus within rows values with different superscript letters differ at P<0.05. Significant P values are highlighted in bold. Gestational intake, GI

(ID) as a random factor. For the leptin profiles, concentrations measured serially from birth were averaged between weeks 0–11, 12–22, 23–38, 40–64, 66–89 and 91–108 (n = 10–11 samples per age block) and prenatal growth status, sex and age were fixed-factors in the model together with all potential interactions. For the NEFA and glycerol concentrations (average fasting levels, slope and AUC after glucose bolus), each sex was examined separately, and prenatal growth-status and age were the fixed-factors. *Post hoc* comparison between groups at all ages was by Fishers LSD method using the SED for comparison of means. Bone mineral density and body fat percentage across the life-course were interrogated using regression analyses: the mean slopes and intercepts and all offspring variables measured on a single occasion at necropsy were analyzed using one-way ANOVA for each sex separately, and by a GLM ANOVA as detailed above to determine the effects of prenatal growth-status, sex, and their potential interaction. Pearson product-moment correlation analysis was used to explore relationships between variables where indicated and data are presented as correlation coefficients (*r*). Statistical significance was taken as $P \leq 0.05$.

## Results

### Experiment 1 (fetal study): Maternal and fetal phenotype

By design the adolescent dams that received a control intake maintained their initial adiposity score from embryo transfer until necropsy at day 130 of gestation (Table 1). By comparison the adiposity score of overnourished dams steadily increased and was equivalent to a gain of 8% body fat by late gestation. Similarly, the overnourished dams had greater gestational weight gain and carcass weight at necropsy, and this anabolic state was reflected by greater plasma insulin and glucose concentrations and attenuated plasma NEFA. Within the overnourished pregnancies, maternal anthropometry and metabolic status was largely independent of whether the fetus was classified as markedly growth-restricted or not: the exception was the maternal NEFA concentrations which were lower in the restricted compared with the non-restricted pregnancies (P<0.05 by unpaired t-test). Irrespective of maternal nutrition or growth category, male fetuses were slightly heavier than females (P = 0.015), and by design growth-restricted fetuses of both sexes were lighter than the corresponding sex of normally-growing control and non-restricted fetuses (32 to 38% lower, P<0.001, Table 2). Relative to the normally-growing control group, average placentome weight was reduced by ~50% in the restricted pregnancies (P<0.001) and although a greater fetal:placental weight ratio implies greater placental efficiency, the higher brain:liver weight ratio is commensurate with brain sparing. Average placentome weight was ~16% lower in the non-restricted overnourished pregnancies relative to the control group (P<0.05) but this degree of placental growth-restriction was below the estimated functional reserve capacity of the ovine placenta [42] and did not impact fetal weight *per se*. Fetal plasma insulin concentrations were normal>non-restricted>restricted for females and males combined, (P = 0.006 by one way ANOVA, normal versus restricted, P<0.05 *post hoc*).

### Fetal adiposity

Absolute PAT mass was positively correlated with fetal weight (r = 0.559, n = 60, P<0.001) and accordingly was lowest in growth-restricted fetuses (restricted <normal = non-restricted for females and males combined, P<0.01 *post hoc*). When expressed relative to fetal weight, PAT mass was influenced by both maternal nutrition/growth category (restricted>normal>non-restricted, P = 0.016) and fetal sex (female>male, P = 0.023, Table 2): the effect of sex was most pronounced in the normally-growing fetuses. Absolute unilocular cell mass was lower in the restricted fetuses (restricted <control = non-restricted for females and males combined,

P<0.05 *post hoc*) but not when expressed relative to fetal weight. Similarly, absolute multilocular cell mass was influenced by maternal nutrition/growth category (restricted<non-restricted<normal, P<0.001) and fetal sex (females>males, P = 0.030). When multilocular cell mass was expressed relative to fetal weight the effect of growth category was largely reversed (restricted = normal >non-restricted, P = 0.007) and the impact of sex maintained (females>males, P = 0.001). All three groups of female fetuses and the control males had a higher relative proportion (volume density) of multilocular compared with unilocular cells (P<0.05 to <0.001) and accordingly a higher absolute and fetal weight-specific multilocular cell mass. Maternal NEFA concentrations at necropsy were positively correlated with multilocular fat cell mass in both female and male fetuses (r = 0.684, n = 35, and r = 0.624, n = 25, respectively, both P<0.001, Fig 1a) while an inverse association was detected between maternal insulin concentrations and the multilocular fat component (females r = -0.629, P<0.001; males r = -0.501, P = 0.011). None of the indices of maternal metabolic status were related to the unilocular fat cell mass (P>0.1). Placental mass was positively associated with both multilocular and unilocular fat cell mass in female fetuses (r = 0.629 and r = 0.632 P<0.001) and with multilocular but not unilocular fat mass in males (r = 0.547, P = 0.005 and r = 0.294, P>0.1). The relationship between the mass of the placenta and the multilocular fat in both sexes is shown in Fig 1b.

## Fetal PAT gene expression

Growth-restricted fetuses had higher relative abundance of *PPARγ*, *G3PDH*, *HSL* and *IGF1-R* mRNA in their PAT than both normal and non-restricted fetuses which were similar to each other (P = 0.007, P<0.001, P = 0.003 and P = 0.019, respectively, Table 3). Consequently, these genes were negatively associated to varying degrees with maternal NEFA concentrations, fetoplacental weights, absolute PAT mass and the multilocular fat component (Table 4). Positive relationships between maternal insulin concentration and both *HSL* and *IGF1-R* mRNA were also evident. The inverse relationship between placental weight and PAT *G3PDH* mRNA expression was pronounced and evident in both female and male fetuses (r = -0.704, P<0.001 and r = -0.562, P = 0.003, Fig 1c). A similar relationship between placentome weight and *UCP-1* mRNA expression was evident (r = -0.728, P<0.001 and r = -0.495, P = 0.012 for females and males, respectively, Fig 1d): the two factor ANOVA confirmed increased abundance of *UCP-1* in restricted fetuses (P<0.001) and the significant growth status x gender interaction (P = 0.009) reflected particularly high expression in growth-restricted females (Table 3). *UCP-1* expression was weakly positively associated with maternal insulin and glucose, and inversely associated with maternal NEFA, absolute fat mass and both the unilocular and multilocular components (Table 4). PAT leptin mRNA was independent of growth status, but females had higher expression than males (Table 3, P = 0.01), commensurate with their greater adiposity. Fetal sex was also the dominant influence on PAT *IGF1* mRNA but in this instance, males had higher expression than females (P = 0.009). *LPL*, *FASN*, Adiponectin, *IGF2* and *IGF2-R* mRNA levels were independent of growth-status and sex but positive relationships between fetal insulin and PAT *LPL* and *FASN* abundance were evident for the study population overall (Table 4) and separately in females (r = 0.468, P = 0.005 and r = 0.445, P = 0.007) and males (r = 0.452, P = 0.027 and r = 0.483, P = 0.017, Fig 1e and 1f).

## Experiment 2 (postnatal study): Phenotype at birth and mid-adulthood

Details of pregnancy outcome and offspring growth, glucose metabolism and body composition at key stages from infancy to mid-adulthood have been presented elsewhere [35]. Of 49 offspring entering the study, 44 completed it and are the focus here. Relative to the normally-growing controls, growth-restricted lambs from overnourished dams were born on average 4

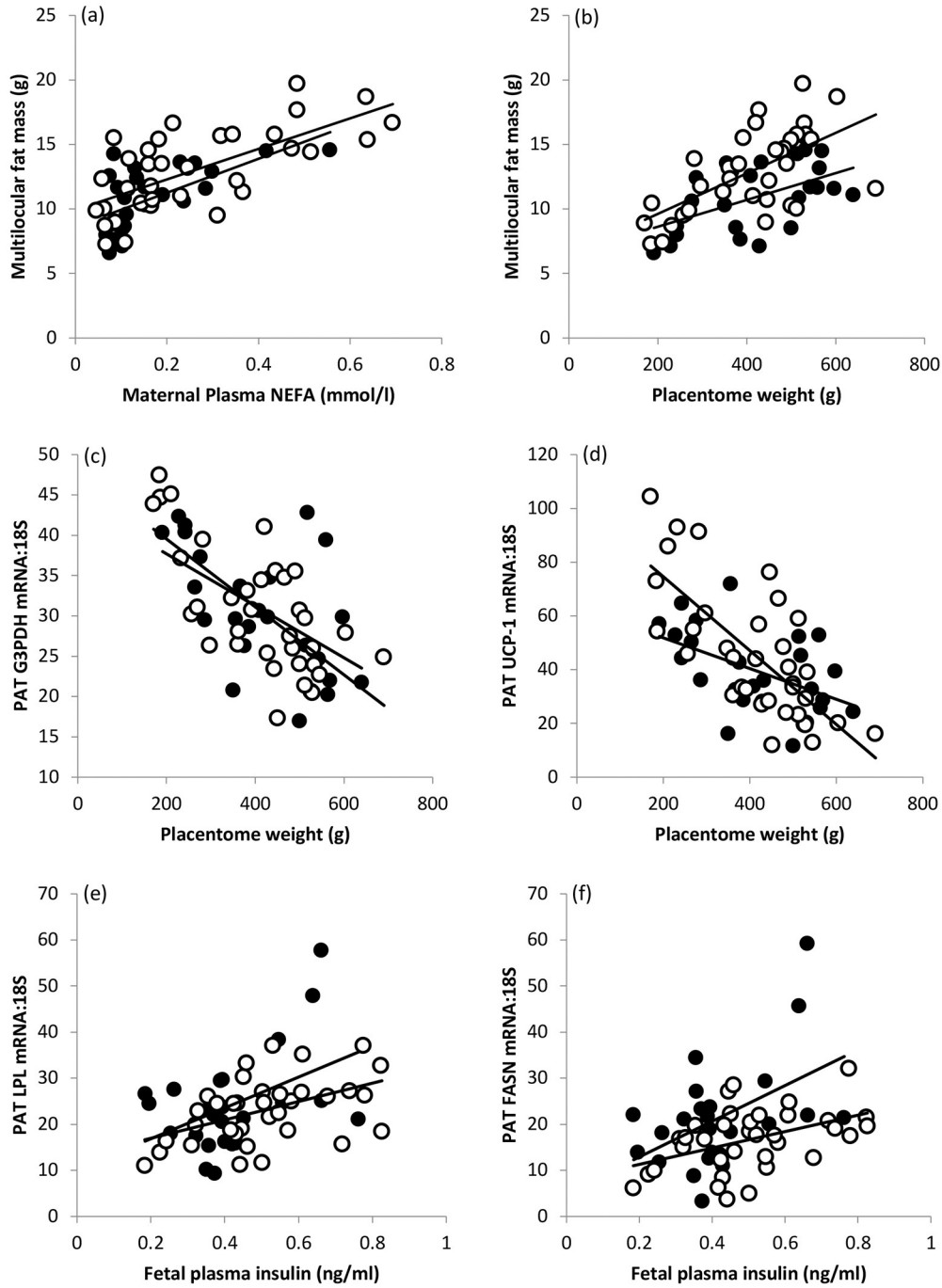

**Fig 1. Indices of nutrient supply associated with fetal fat phenotype and gene expression.** Relationships between (a) maternal plasma NEFA concentration and fetal multilocular fat cell mass, (b) placentome weight and multilocular fat cell mass, (c) placentome weight and fetal perirenal adipose tissue (PAT) *G3PDH* mRNA, (d) placentome weight and PAT *UCP-1* mRNA, (e) fetal plasma insulin concentration and PAT *LPL* mRNA and (f) fetal plasma insulin and PAT *FASN* mRNA at day 130 of gestation. For female (open circles) and male (closed circles) fetuses respectively in (a) r = 0.684, P<0.001 and r = 0.624, P = 0.001; in (b) r = 0.629, P<0.001 and r = 0.547, P = 0.005; in (c) r = -0.704, P<0.001 and r = -0.562, P = 0.003; in (d) r = -0.728, P<0.001 and r = -0.495, P = 0.012; in (e) r = 0.468, P = 0.005 and r = 0.452, P = 0.027; in (f) r = 0.445, P = 0.007 and r = 0.483, P = 0.017.

**Table 4. Relationship between perirenal fat gene expression (relative to 18S) and maternal metabolic status, fetal plasma insulin, conceptus mass and fat mass at day 130 of gestation, irrespective of maternal nutrition, prenatal growth category and sex (Experiment 1).**

| | PPARγ | G3PDH | LPL | FASN | HSL | Leptin | Adiponectin | IGF1 | IGF2 | IGF1-R | IGF2-R | UCP1 |
|---|---|---|---|---|---|---|---|---|---|---|---|---|
| Maternal glucose | 0.239 | 0.190 | 0.097 | 0.094 | 0.176 | -0.131 | -0.192 | 0.108 | 0.171 | **0.277*** | 0.247 | **0.290*** |
| Maternal NEFA | **-0.337**** | **-0.324*** | -0.201 | -0.097 | **-0.469**** | -0.024 | 0.100 | -0.003 | -0.148 | **-0.366**** | **-0.346**** | **-0.330**** |
| Maternal insulin | 0.180 | **0.264*** | -0.064 | -0.078 | **0.435**** | 0.060 | 0.128 | -0.179 | -0.104 | **0.303**** | **0.408**** | **0.322*** |
| Placentome wt, g | **-0.335**** | **-0.644**** | 0.233 | **0.258*** | **-0.416**** | -0.170 | -0.061 | **0.306*** | 0.241 | -0.212 | -0.197 | **-0.633**** |
| Fetal wt, g | **-0.297*** | **-0.505**** | 0.052 | 0.093 | **-0.329**** | -0.198 | -0.086 | **0.297*** | 0.179 | -0.155 | -0.157 | **-0.451**** |
| Fetal insulin | -0.046 | **-0.289*** | **0.410**** | **0.346**** | -0.180 | 0.256 | 0.012 | 0.233 | 0.239 | -0.144 | -0.098 | -0.181 |
| Perirenal fat g | **-0.347**** | **-0.480**** | -0.009 | 0.015 | **-0.404**** | 0.106 | 0.123 | -0.011 | -0.078 | **-0.441**** | **-0.366**** | **-0.434**** |
| Unilocular fat mass, g | -0.144 | **-0.344**** | 0.050 | 0.125 | -0.190 | 0.090 | 0.115 | 0.017 | 0.059 | -0.210 | -0.117 | **-0.335**** |
| Multilocular fat mass, g | **-0.404**** | **-0.439**** | -0.056 | -0.081 | **-0.453**** | 0.086 | 0.089 | -0.032 | -0.166 | **-0.491**** | **-0.411**** | **-0.376**** |

Significant *r* values highlighted in bold

*$P < 0.05$,

**$P < 0.01$,

***$P < 0.001$.

days earlier (mean±sem: 143.8±0.27 versus 139.8±0.42 days), were ~42% lighter at birth (5578 ±141 versus 3234±167g, Table 5), had a smaller girth at the umbilicus (40.4±0.51 versus 34.2 ±0.71cm), and a reduced height at the shoulder (34.0±0.40 versus 27.4±0.76cm). This reduced prenatal growth reflected a 43% decrease in fetal placental weight (523±29 versus 297±20g) and all these measures of size at birth were statistically significant (P<0.001) and independent of offspring sex.

## Plasma leptin and offspring adiposity

Weekly changes in peripheral plasma leptin concentrations during the suckling phase and concentrations averaged during defined ages thereafter are shown in Fig 2a and 2b. Neither prenatal growth status nor sex influenced leptin concentrations at birth or 1 week of age; thereafter levels diverged with females greater than males throughout the rest of the suckling period (P<0.001). Indeed, female offspring had persistently higher leptin concentrations than males throughout the life-course (P<0.001). In both sexes average leptin secretion increased between the 12 to 22, and, 23 to 38 weeks of age periods: maximum concentrations in females were reached between 66 and 89 weeks and plateaued thereafter, whereas in males the maximum levels observed were between weeks 91 and 108 of age. Although the repeated measures analysis did not detect an influence of prenatal growth category overall, there was a prenatal growth * age interaction (P = 0.042) with *post hoc* analysis indicating that growth-restricted males had a trend for higher leptin concentrations than normal males from adolescence forwards. Irrespective of prenatal growth status and sex, average peripheral leptin concentrations in the age period preceding DEXA assessment were positively associated with percentage body fat at 11, 41 and 64 weeks (Fig 2c, 2d and 2e, P<0.001) but by 107 weeks the relationship was no longer significant (Fig 2f, P>0.08). Similarly, plasma leptin concentrations immediately pre-necropsy were equivalent between the four groups. In contrast there was a trend for lower plasma adiponectin concentrations pre-necropsy in growth-restricted versus normal birthweight males (P = 0.053) and the plasma adiponectin: leptin ratio, a putative marker of insulin resistance [43], was reduced (P = 0.04) in the former group. Birth weight and plasma adiponectin at necropsy was positively associated in males (r = 0.496, P = 0.036) but not females.

**Table 5. Phenotype, plasma adipokines and perirenal fat gene expression at necropsy in mid-adult life in relation to gestational intake, prenatal growth status and sex (Experiment 2).**

| Gestational intake | Control | Overnourished | | Control | Overnourished | | P value glm | | |
|---|---|---|---|---|---|---|---|---|---|
| Prenatal growth status | Normal | Restricted | | Normal | Restricted | | Prenatal growth status | Sex | Interaction |
| Gender | Female, n = 10 | Female, n = 16 | NF vs RF | Male, n = 11 | Male, n = 7 | NM vs RM | | | |
| Reference birth weight, g | 5287±251[a] | 3361±201[b] | **<0.001** | 5843±98[a] | 2943±287[b] | **<0.001** | **<0.001** | 0.634 | **0.027** |
| Age at necropsy, d | 768±0.8[a] | 767±0.7[a] | 0.373 | 774±0.4[b] | 773±0.8[b] | 0.150 | 0.128 | **<0.001** | 0.848 |
| Live weight (LW), kg | 109.5±1.62[a] | 104.92±2.64[a] | 0.256 | 147.3±1.31[b] | 137.9±4.67[b] | **0.033** | **0.014** | **<0.001** | 0.387 |
| ¥Bone mineral density, g/cm | 1.026±0.018[a] | 0.968±0.014[a] | **0.018** | 1.373±0.027[b] | 1.259±0.057[c] | 0.061 | **0.003** | **<0.001** | 0.306 |
| ¥Body fat % | 44.10±1.209[a] | 52.02±1.078[b] | **0.001** | 43.72±0.941[a] | 46.19±0.832[a] | 0.088 | **<0.001** | **0.010** | **0.023** |
| ¥Fat:lean mass | 0.841±0.045[b] | 1.142±0.056[a] | **0.001** | 0.821±0.035[b] | 0.902±0.031[b] | 0.128 | **0.001** | **0.016** | **0.040** |
| Bone mineral density vs age | | | | | | | | | |
| Average slope, g/cm per week | 0.0043±0.0003[a] | 0.0043±0.0002[a] | 0.924 | 0.0081±0.0003[b] | 0.0073±0.0005[b] | 0.205 | 0.191 | **<0.001** | 0.224 |
| Average intercept, g/cm | 0.611±0.016[a] | 0.546±0.014[b] | **0.006** | 0.576±0.015[ab] | 0.536±0.020[b] | 0.121 | **0.003** | 0.179 | 0.452 |
| Body fat % vs age | | | | | | | | | |
| Average slope, % per week | 0.2282±0.0146[a] | 0.2837±0.0151[b] | **0.020** | 0.2931±0.0135[b] | 0.2776±0.0118[ab] | 0.434 | 0.210 | 0.069 | **0.029** |
| Average intercept, % | 21.16±0.826[a] | 21.77±1.180[a] | 0.712 | 11.36±0.916[a] | 14.12±1.180[b] | 0.081 | 0.155 | **<0.001** | 0.362 |
| Perirenal fat mass, g | 1713±232[a] | 2489±176[c] | **0.014** | 3453±221[b] | 2797±210[bc] | 0.061 | 0.785 | **<0.001** | **0.002** |
| Perirenal fat mass, g/kg LW | 15.5±2.04[a] | 23.7±1.73[b] | **0.007** | 23.4±1.47[b] | 20.2±1.23[ab] | 0.146 | 0.185 | 0.232 | **0.004** |
| Plasma leptin, ng/ml | 25.3±1.20 | 28.1±1.64 | 0.247 | 23.8±2.07 | 26.0±0.46 | 0.417 | 0.167 | 0.314 | 0.878 |
| Plasma adiponectin, μg/ml | 4.20±0.190 | 4.66±0.267 | 0.241 | 4.61±0.275 | 3.78±0.239 | 0.053 | 0.506 | 0.400 | **0.026** |
| Plasma adiponectin: leptin | 172±9.1 | 182±19.3 | 0.689 | 206±20.1 | 146±10.9 | **0.040** | 0.965 | 0.200 | 0.072 |
| 18s | 0.035±0.003 | 0.033±0.001 | 0.314 | 0.033±0.002 | 0.033±0.002 | 0.962 | 0.495 | 0.907 | 0.544 |
| $PPAR\gamma$:18s | 17.40±1.30[c] | 23.0±1.29[bc] | **0.009** | 29.14±2.12[ab] | 30.91±2.81[a] | 0.617 | 0.057 | **<0.001** | 0.314 |
| G3PDH:18s | 29.02±2.28[a] | 28.22±2.20[a] | 0.815 | 22.72±1.62[ab] | 17.97±1.41[b] | 0.059 | 0.212 | **0.001** | 0.370 |
| LPL:18s | 15.878±0.732[a] | 17.924±0.994[ab] | 0.161 | 20.519±0.877[b] | 20.164±1.580[ab] | 0.834 | 0.441 | **0.003** | 0.276 |
| FASN:18s | 11.03±1.99[c] | 14.17±1.71[bc] | 0.257 | 21.3±2.51[ab] | 25.25±4.24[a] | 0.403 | 0.168 | **<0.001** | 0.873 |
| HSL:18s | 10.53±1.33[c] | 19.14±2.12[bc] | **0.008** | 38.21±5.62[a] | 32.19±2.17[ab] | 0.416 | 0.730 | **<0.001** | **0.047** |
| Leptin:18s | 12.124±1.874[a] | 19.484±1.785[ab] | **0.013** | 24.297±3.038[b] | 26.398±2.637[b] | 0.637 | 0.062 | **<0.001** | 0.293 |
| Adiponectin:18s | 10.378±1.510[a] | 15.476±1.340[a] | **0.023** | 26.989±2.801[b] | 24.029±2.121[b] | 0.461 | 0.610 | **<0.001** | 0.060 |
| IGF1:18s | 26.76±2.37 | 25.34±1.96 | 0.653 | 27.52±1.98 | 22.25±1.43 | 0.074 | 0.130 | **0.594** | 0.380 |
| IGF2:18s | 27.65±4.12 | 25.62±4.12 | 0.645 | 20.82±2.18 | 17.43±2.08 | 0.306 | 0.350 | **0.013** | 0.814 |
| INSR:18s | 17.046±1.060 | 19.457±0.768 | 0.074 | 17.722±1.192 | 19.632±1.486 | 0.332 | 0.059 | 0.703 | 0.822 |

¥By DEXA at 107 weeks. Normal female (NF), restricted female (RF), normal male (NM), restricted male (RM)

## NEFA and glycerol: Fasting concentrations and metabolism during glucose tolerance tests

Summary NEFA and glycerol concentrations in the fasted state and following glucose challenge at 7, 32, 60, 85 and 106 weeks of age in relation to prenatal growth status and sex are presented in Table 6, and the latter acted as a proxy measure of insulin-induced adipose tissue lipid metabolism. NEFA and glycerol profiles following the glucose challenge close to study end are shown in Fig 3a and 3b. The mixed-method repeated-measures analysis approach indicates that over the five life-stages, fasting NEFA concentrations were significantly influenced by prenatal

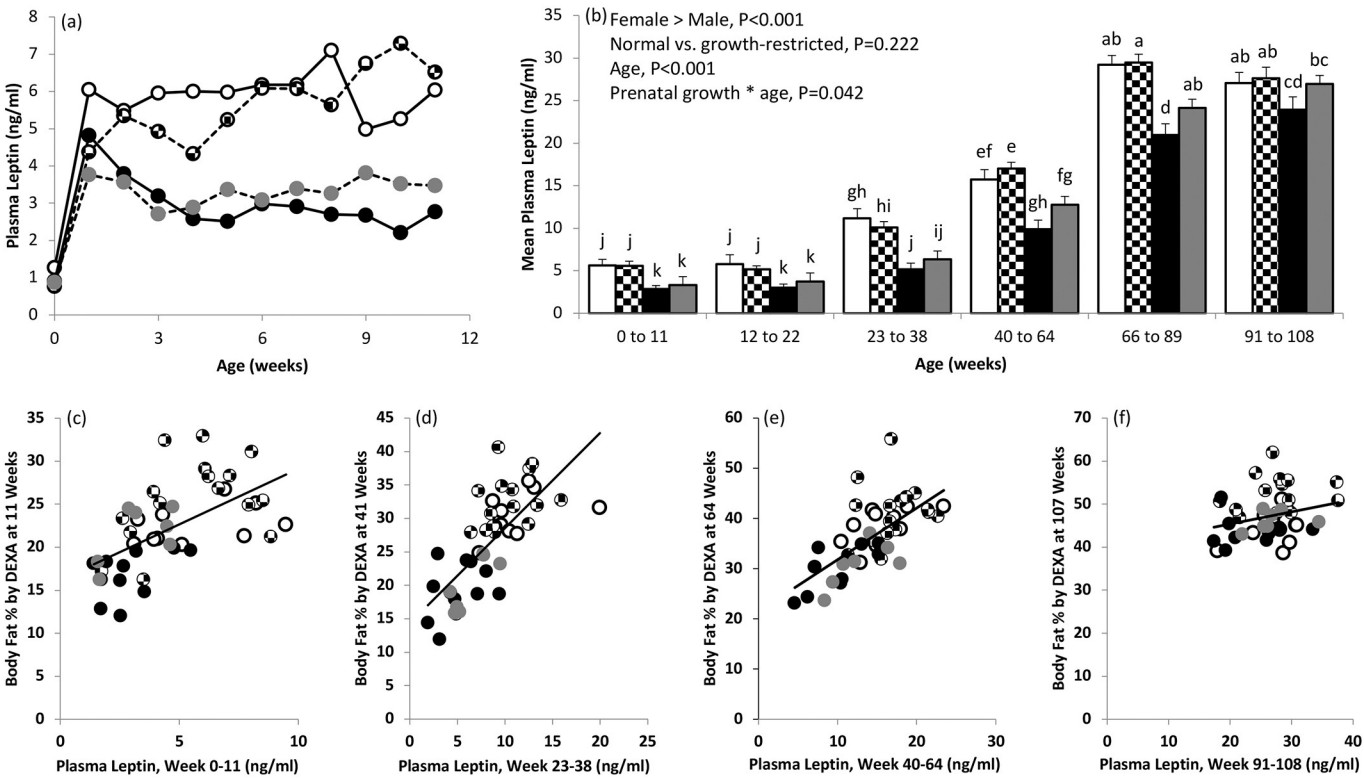

**Fig 2. Circulating leptin and body fat percentage across the life-course.** Effect of prenatal growth-restriction and sex on peripheral plasma leptin concentrations from birth to weaning at 11 weeks of age (a) and on average leptin concentrations during defined periods throughout the life-course (b) in offspring that completed the study (n = 44). Relationship between average leptin concentrations (mean of 10–11 samples) between age 0 to 11(c), 23 to 38 (d), 40 to 64 (e) and 91 to 108 (f) weeks and body fat percentage measured by DEXA at 11, 41, 64 and 107 weeks of age. Normal birthweight females are denoted by open bars or circles; growth-restricted females by checked bars or circles; normal males by black bars or circles and growth-restricted males by grey bars or circles. In (b) leptin concentrations were influenced by age and sex and there was a significant prenatal growth * age interaction. *Post hoc* comparisons were used to further differentiate between the four groups across the life-course and bars with different superscript letters differ at P<0.05. Irrespective of prenatal growth category and gender in (c) r = 0.592, P<0.001; in (d) r = 0.732, P<0.001; in (e) r = 0.683, P<0.001 and in (f) r = 0.260, P = 0.088, not significant.

growth-status and postnatal age in both sexes with *post hoc* analysis revealing markedly higher levels in restricted compared with normal females at 32, 60 and 106 weeks, and in restricted versus normal males at 32 and106 weeks. Irrespective of prenatal growth category, males had higher fasting NEFA concentrations than females at 7, 32 and 60 weeks but by study end the sexes were equivalent. NEFA concentrations increased transiently after glucose administration before falling sharply in response to the associated increase in insulin (see [35] for insulin pro-files) and concentrations did not return to baseline within the sampling period at the 60, 85 or 106 week-challenges. The insulin-induced fall in NEFA output during the glucose tolerance test was reduced in growth-restricted versus normal offspring of both sexes, resulting in a higher NEFA AUC: the *post hoc* analysis indicated that the most pronounced differences were at 32, 60 and 106 weeks in females and at 7, 32, 60 and 106 weeks in males. These differences were largely independent of any change in slope from 10 to 30 minutes after glucose bolus. Birthweight was inversely correlated with fractional growth rate (FGR) for weight during the suckling phase (D0-77) in both sexes (females r = -0.898, P<0.001; males r = -0.924, P<0.001) and strong associa-tions between birthweight, early relative growth and NEFA concentrations were evident close to study end. Thus, baseline and AUC NEFA decreased with increasing birthweight in both sexes (Fig 3c and 3e) while baseline (not shown) and AUC NEFA at 106 weeks of age were positively associated with FGR during early postnatal life (Fig 3f). In contrast there was no relationship

**Table 6. NEFA and glycerol concentrations in the fasted state and following glucose challenge at 7, 32, 60, 85 and 106 weeks of age in relation to gestational intake, prenatal growth status and sex (Experiment 2).**

| Gestational intake | Control | | Overnourished | | P value (glm) | | |
|---|---|---|---|---|---|---|---|
| Prenatal growth status | Normal | Restricted | Normal | Restricted | Prenatal growth status | Sex | Interaction |
| Gender | Female, n = 10 | Female, n = 16 | Male, n = 11 | Male, n = 7 | | | |
| Fasting NEFA, mmol/l | | | | | | | |
| - 7 weeks | 0.179±0.013[f] | 0.211±0.030[f] | 0.201±0.030[f] | 0.362±0.070[ef] | **0.012** | **0.023** | 0.086 |
| - 32 weeks | 0.789±0.058[bc] | 1.064±0.072[a] | 1.025±0.081[bc] | 1.314±0.069[a] | **0.001** | **0.004** | 0.926 |
| - 60 weeks | 0.696±0.028[cd] | 0.863±0.048[b] | 0.910±0.084[bc] | 1.062±0.084[b] | **0.019** | **0.003** | 0.912 |
| - 85 weeks | 0.487±0.049[e] | 0.605±0.041[de] | 0.404±0.040[e] | 0.447±0.054[de] | 0.101 | **0.016** | 0.444 |
| - 106 weeks | 0.556±0.039[e] | 0.861±0.053[b] | 0.622±0.046[d] | 0.860±0.100[c] | **<0.001** | 0.602 | 0.584 |
| [¥]RM P value - growth status | | **<0.001** | | **0.016** | | | |
| - age | | **<0.001** | | **<0.001** | | | |
| - growth status x age | | **0.020** | | 0.281 | | | |
| NEFA slope, 10 to 30min after glucose (µmol/l per min) | | | | | | | |
| - 7 weeks | -3.7±0.98[a] | -4.75±1.18[a] | -6.0±2.16[a] | -9.8±2.17[a] | 0.252 | 0.080 | 0.522 |
| - 32 weeks | -23.7±3.0[d] | -34.1±2.45[e] | -29.6±4.1[c] | -30.0±5.86[c] | 0.153 | 0.817 | 0.185 |
| - 60 weeks | -13.2±0.94[bc] | -19.7±1.81[cd] | -19.0±1.48[b] | -19.2±4.53[b] | 0.141 | 0.234 | 0.154 |
| - 85 weeks | -13.3±1.55[bc] | -11.6±1.51[b] | -12.5±1.53[ab] | -8.0±1.71[a] | 0.075 | 0.201 | 0.412 |
| - 106 weeks | -14.5±5.76[bc] | -8.80±2.84[ab] | -12.5±2.38[ab] | -9.3±2.17[a] | 0.227 | 0.829 | 0.727 |
| [¥]RM P value - growth status | | 0.285 | | 0.786 | | | |
| - age | | **<0.001** | | **<0.001** | | | |
| - growth status x age | | **0.005** | | 0.660 | | | |
| NEFA AUC (mmol/l x min) | | | | | | | |
| - 7 weeks | 11.93±0.936[f] | 14.01±1.80[ef] | 12.84±2.49[ef] | 23.34±5.02[bcd] | **0.017** | **0.050** | 0.104 |
| - 32 weeks | 19.55±2.04[cd] | 28.71±2.02[a] | 23.26±1.66[bc] | 32.30±4.84[a] | **0.001** | 0.161 | 0.982 |
| - 60 weeks | 14.29±0.910[ef] | 23.56±1.382[bc] | 16.30±1.34[de] | 34.7±6.78[a] | **<0.001** | **0.015** | 0.084 |
| - 85 weeks | 15.30±1.28[def] | 17.11±1.54[de] | 9.84±1.28[f] | 13.78±3.11[ef] | 0.117 | **0.019** | 0.557 |
| - 106 weeks | 18.86±1.90[cde] | 26.70±1.77[ab] | 18.42±2.18[cde] | 27.84±3.20[ab] | **<0.001** | 0.877 | 0.727 |
| [¥]RM P value - growth status | | **<0.001** | | **0.006** | | | |
| - age | | **<0.001** | | **<0.001** | | | |
| - growth status x age | | **0.049** | | 0.063 | | | |
| Fasting glycerol, µmol/l | | | | | | | |
| - 7 weeks | 61.5±3.33[f] | 80.2±8.86[def] | 63.5±6.28[de] | 75.4±8.21[cd] | 0.061 | 0.863 | 0.668 |
| - 32 weeks | 97.4±8.95[bcd] | 128.7±8.12[a] | 103.0±5.67[b] | 126.7±9.71[a] | **0.003** | 0.835 | 0.663 |
| - 60 weeks | 93.9±7.55[cde] | 107.6±5.85[bc] | 91.9±7.60[bc] | 109.5±10.51[ab] | **0.050** | 0.994 | 0.801 |
| - 85 weeks | 71.6±3.79[ef] | 84.7±8.45[def] | 52.3±4.37[e] | 59.1±4.32[de] | 0.177 | **0.004** | 0.664 |
| - 106 weeks | 109.5±6.49[abc] | 117.7±11.2[ab] | 69.4±3.26[d] | 91.5±7.44[bc] | 0.118 | **0.001** | 0.465 |
| [¥]RM P value - growth status | | **0.031** | | **0.011** | | | |
| - age | | **<0.001** | | **<0.001** | | | |
| - growth status x age | | 0.582 | | 0.628 | | | |
| Glycerol slope, 10-30min after glucose (µmol/l per min) | | | | | | | |
| - 7 weeks | -1.03±0.331[a] | -1.46±0.333[a] | -0.88±0.236[a] | -1.19±0.468[a] | 0.305 | 0.552 | 0.854 |
| - 32 weeks | -1.62±0.296[ab] | -2.94±0.561[bc] | -1.22±0.305[a] | -1.76±0.435[a] | 0.071 | 0.128 | 0.444 |
| - 60 weeks | -1.35±0.194[a] | -3.06±0.335[c] | -1.67±0.190[a] | -3.54±0.596[b] | **<0.001** | 0.255 | 0.824 |

*(Continued)*

**Table 6.** (Continued)

| Gestational intake | Control | | Overnourished | | P value (glm) | | |
|---|---|---|---|---|---|---|---|
| Prenatal growth status | Normal | Restricted | Normal | Restricted | Prenatal growth status | Sex | Interaction |
| Gender | Female, n = 10 | Female, n = 16 | Male, n = 11 | Male, n = 7 | | | |
|  - 85 weeks | -2.26±0.473ᵃᵇ | -3.03±0.477ᶜ | -1.03±0.374ᵃ | -1.69±0.407ᵃ | 0.155 | **0.013** | 0.906 |
|  - 106 weeks | -2.97±1.05ᵇᶜ | -3.43±0.376ᶜ | -3.25±0.531ᵇ | -3.60±0.594ᵇ | 0.541 | 0.730 | 0.931 |
| ¥RM P value - growth status | | **0.027** | | **0.021** | | | |
|  - age | | **0.001** | | **<0.001** | | | |
|  - growth status x age | | 0.510 | | 0.300 | | | |
| Glycerol AUC (µmol/l x min) | | | | | | | |
|  - 7 weeks | 4255±331ᵇᶜ | 5390±508ᵃ | 4176±412ᵇ | 5098±658ᵃ | **0.050** | 0.717 | 0.836 |
|  - 32 weeks | 2432±233ᵉ | 3654±260ᶜᵈ | 2576±194ᶜᵈᵉ | 3209±351ᶜ | **0.002** | 0.588 | 0.290 |
|  - 60 weeks | 2588±253ᵉ | 3235±320ᵈᵉ | 1860±110ᵉᶠ | 3026±462ᶜ | **0.007** | 0.148 | 0.419 |
|  - 85 weeks | 2724±257ᵈᵉ | 3102±234ᵈᵉ | 1738±163ᶠ | 2070±203ᵈᵉᶠ | 0.151 | **<0.001** | 0.926 |
|  - 106 weeks | 4873±379ᵃᵇ | 4514±300ᵇ | 2577±205ᶜᵈᵉ | 2856±212ᶜᵈ | 0.898 | **<0.001** | 0.312 |
| ¥RM P value - growth status | | **0.034** | | **0.008** | | | |
|  - age | | **<0.001** | | **<0.001** | | | |
|  - growth status x age | | 0.093 | | 0.545 | | | |

¥Repeated measures ANOVA carried out separately for females and males, with *post hoc* comparisons using Fisher's LSD method. Values with a different superscript letter differ <0.05. Significant P values are highlighted in bold.

between current liveweight at 106 weeks and the NEFA response to glucose challenge in either sex (r = -0.220, P>0.3 and r = -0.205, P>0.4).

Fasting glycerol concentrations were also moderately impacted by prenatal growth category. Relative to the normally growing comparators the most distinct elevations in fasting concentrations were detected at 32 and 106 weeks in restricted males and at 32 weeks in restricted females. However, the GLM approach indicates that offspring sex is the dominant influence during the second year of life with females having higher concentrations than males at 85 and 106 weeks, reflecting their greater adiposity. Following glucose administration, the fall in glycerol concentrations (slope) was modestly influenced by prenatal growth category in both sexes and was particularly steep in growth-restricted females at 60 and 95 weeks, and in growth-restricted males at 60 weeks of age. As above the reduced insulin sensitivity during the glucose tolerance test as a whole resulted in a higher AUC glycerol in growth-restricted versus normal offspring of both sexes with the most distinct differences observed at challenges carried out between 7 and 60 weeks of age. Offspring sex (female>male) was the overriding influence on AUC glycerol at 85 and 106 weeks. Irrespective of prenatal growth status or sex, circulating NEFA and glycerol concentrations (fasting and post-glucose challenge) were positively associated at all 5 life stages (ranging from r = 0.471 to 0.801, P = 0.002 to <0.001). Close to study end birthweight was inversely associated with fasting glycerol concentrations in males but not females (Fig 3d).

## Offspring phenotype in relation to PAT gene expression at necropsy

As reported previously [35] and included here for comparative purposes bone mineral density and live weight at study end in mid-adult life were influenced by prenatal growth-status (restricted< normal) and gender (females <males, Table 5), while in contrast percentage body fat and fat:lean mass were greater in growth-restricted offspring and in females compared with males, predominantly due to higher adiposity in the growth-restricted female

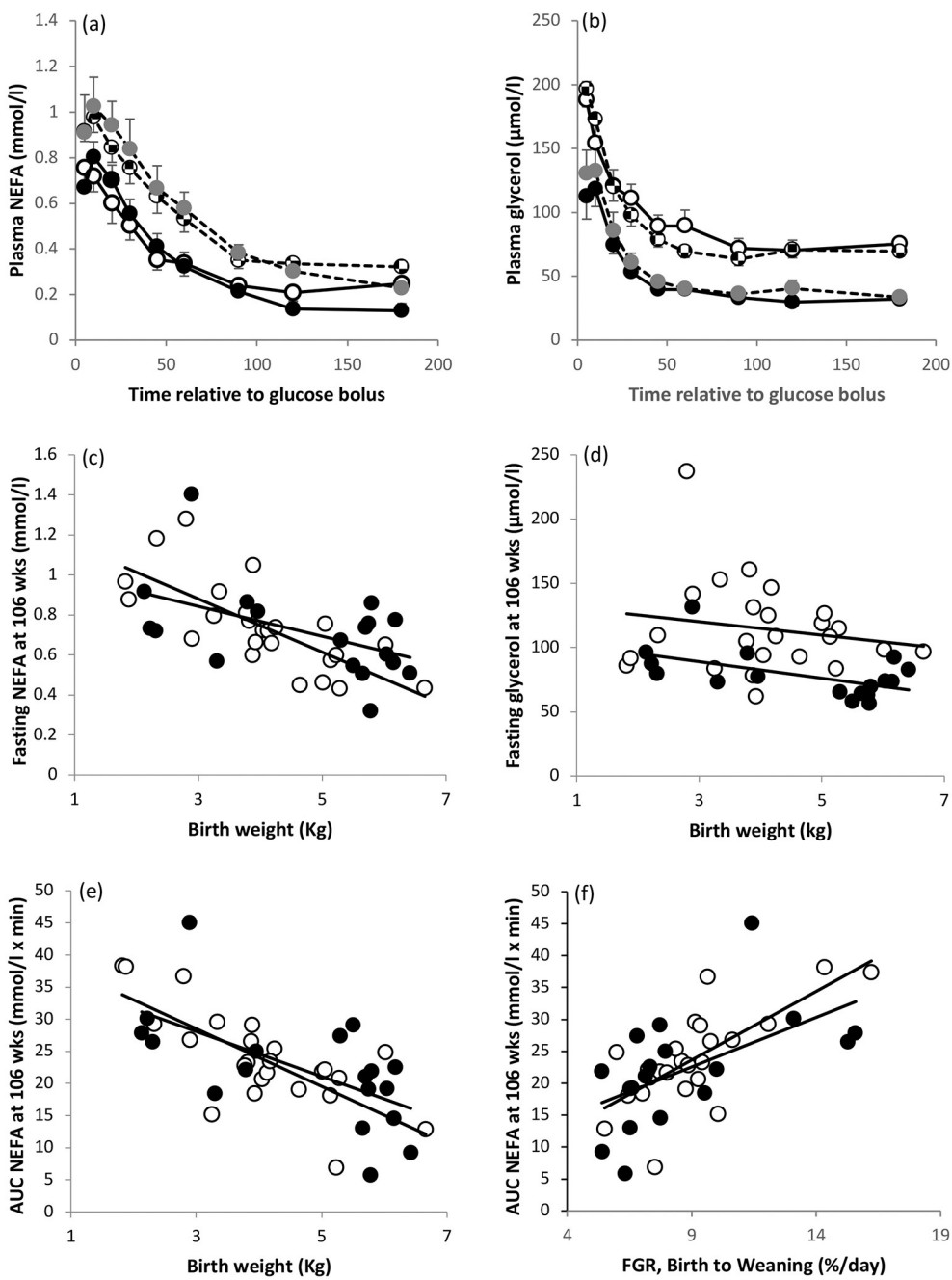

**Fig 3. Lipid metabolism in mid-adult life in relation to early growth trajectories and sex.** Plasma concentrations of NEFA (a) and glycerol (b) following bolus administration of glucose at 106 weeks of age in normal (open circle) and prenatally growth-restricted (checked circle) females, and normal (black circle) and growth- restricted (grey circle) males. Values are mean ± sem. Relationship between birthweight and fasting concentrations of NEFA (c) and fasting glycerol (d) at 106 weeks of age, and relationship between birthweight (e) and fractional growth rate (FGR) from birth to weaning (f) and NEFA AUC at 106 weeks of age. For female (open circles) and male (solid circles) offspring respectively in (c) r = -0.756, P<0.001 and r = -0.500, P = 0.034; in (d) r = -0.234, P>0.2 and r = -0.548, P = 0.019; in (e) r = -0.726, P<0.001 and r = -0.611, P = 0.007; in (f) r = 0.712, P<0.001 and r = 0.577, P = 0.016.

group. Herein we have additionally used regression analysis of the serial DEXA data to estimate body composition at birth (0 weeks, intercept) and the average change across the life-course thereafter (slope). Individual regression analysis of bone mineral density against age revealed that the average slope of the regression lines was ~80% steeper in males than in females (P<0.001), independent of prenatal growth. Conversely the mean intercept (estimated BMD at birth) was ~9% lower in growth-restricted compared with normal groups (P = 0.003), irrespective of sex. A prenatal growth x sex interaction (P = 0.029) was evident for the average slope of the individual regression lines of body fat percentage against age. This reflected both a relatively higher rate of body fat accrual in (a) normal males versus normal females, and in (b) growth-restricted females versus normal females (28 and 24%, respectively, both P<0.05, from *post hoc* analysis). In contrast the mean intercept (estimated body fat % at birth) was ~ 73% higher in females than males (P<0.001), irrespective of prenatal growth category. DEXA is considered the gold standard for assessing body composition and the average slopes of the individual regression lines for fat accrual across the life course were wholly mirrored by corresponding differences in absolute and live weight-specific perirenal adipose tissue mass at necropsy. A prenatal growth x sex interaction in absolute and relative PAT mass was again evident (P = 0.002 and P = 0.004, respectively) and on a live-weight specific basis reflected more renal fat in normal males versus normal females (51%), and in growth-restricted females versus normal females (53%). Moreover, the individual slopes of the regression lines for fat % against age were positively associated with absolute and relative PAT mass for the population overall (r = 0.728 and 0.630, P<0.001) and in females (r = 0.775 and 0.791, P<0.001) but not males (P>0.1) separately.

Offspring sex was the dominant influence on PAT gene expression per unit tissue at necropsy with the abundance of *PPARɤ*, *LPL*, *FASN*, *HSL*, leptin and adiponectin mRNA being robustly greater, and *G3PDH* and *IGF2* mRNA lower in males compared with females (P = 0.013 to <0.001, Table 5). When the sexes were considered separately none of the PAT genes were differentially expressed in normal versus growth-restricted male offspring. However, growth-restricted females were characterised by higher abundance of *PPARɤ*, *HSL*, leptin and adiponectin mRNA than normal females (P = 0.023 to P = 0.009) in keeping with their greater relative adiposity at study end.

## Discussion

These studies in a robust and highly controlled model of prenatal growth-restriction demonstrate that impaired fetal nutrient supply and sex both influence prenatal adipose tissue development and selected gene expression with consequences for lipid metabolism and body composition persisting into adult life.

### Fetal growth, adiposity and perirenal fat gene expression

In the prenatal study, comparing growth-restricted and non-restricted fetuses gestated by over-nourished adolescent dams with the normally growing controls allowed us to differentiate between maternal nutrition and placental nutrient supply. It is clear that when placental size was impaired beyond its functional reserve capacity in rapidly growing overnourished dams that the resulting reduction in placental nutrient transport limited growth of the fetal body, total perirenal fat deposition, and mass of its unilocular and multilocular components. However, when expressed relative to fetal weight, the mass of this dominant prenatal fat depot, and in particular the proportion of multilocular adipocytes, was increased. Similarly, the expression of *UCP1*, a key marker of brown adipose tissue functionality, was enhanced. These *in utero* adaptations in nutrient partitioning priorities are likely to be beneficial to neonatal

thermogenesis and survival since these small fetuses have a relatively high surface area to volume ratio making them particularly vulnerable to hypothermia, and they are also destined to be born prematurely. Indeed, the up-regulated *UCP1* expression is commensurate with an early preparation for thermogenesis [11] and anecdotally these growth-restricted fetuses which are viable following spontaneous delivery at day 135 of gestation onwards exhibit normal vigour immediately after birth, suggesting that thermogenesis is largely unperturbed. The mechanistic basis of the increase in relative adiposity in late gestation may begin earlier in pregnancy during the proliferative phase of adipocyte development when both maternal and fetal glucose concentrations in overnourished dams are high, and before placental size *per se* is constrained [44]. Although direct evidence to support such a hypothesis in the present *overnourished model* is not available, we have previously shown that the lean fetal phenotype that characterises the contrasting *undernourished model* emerges at mid-gestation when the expression of genes involved in adipose tissue proliferation and differentiation, namely *PPARγ*, *IGF1*, *IGF2* and their receptors, were already attenuated by maternal undernutrition and the associated low glucose supply [22]. In addition to glucose, evidence is beginning to accumulate that maternal lipids (triglycerides and free fatty acids) are important substrates for fetal fat accretion and neonatal adiposity in human pregnancies characterised by nutrient excess at conception and fetal overgrowth [45–47]. At first glance a positive correlation between circulating maternal NEFA concentrations and fetal multilocular cell mass in the present study appears to support this concept of 'making fat from fat' but no relationship was evident between maternal NEFA and unilocular (fat storage) cell mass. However, in contrast to the aforementioned humans with pre-pregnancy obesity, the *overnourished* adolescent sheep mothers are anabolic throughout, and continue to grow and progressively accrue body fat as pregnancy advances. Thus, circulating fatty acid concentrations remain low throughout gestation [36], particularly in the dams gestating the most growth-restricted fetuses with the relatively fat phenotype (present study).

Irrespective of substrate, the relatively fat phenotype of the fetuses defined as growth-restricted by late gestation was associated with greater adipose tissue expression of the primary driver of adipogenesis, namely *PPARγ*, and specific genes involved in lipogenesis and fat metabolism, namely *G3PDH* and *HSL*, respectively. The negative association between placental size and the expression of these three genes, in addition to *UCP1*, implies they are a sensitive barometer of impaired fetal nutrient supply and play a role in helping the growth-restricted fetus adapt in preparation for the energetic demands of extrauterine life. This contrasts somewhat with prior data in adult sheep pregnancies demonstrating a link between a nutritionally-mediated increase in maternal and hence fetal glucose during late gestation and greater fetal PAT *PPARγ* gene expression, albeit independent of any change in fetal weight or adipose tissue mass [17]. In the latter study, expression of other lipogenic (*LPL*) and adipokine (adiponectin and leptin) genes were also greater in the overfed group, and this was replicated for *LPL* and adiponectin by intrafetal administration of the *PPARγ* agonist, rosiglitazone [48]. This led these researchers to suggest that precocial activation of these genes, with increased *PPARγ* as the initiating mechanism prenatally, could underlie the increased subcutaneous adiposity observed separately in the offspring of overfed dams at 30 days postnatal [49]. While we did detect the expected positive relationship between fetal insulin concentrations and both *LPL* and *FASN* abundance in PAT for the population as a whole herein, the expression of *LPL*, *FASN*, adiponectin and leptin were independent of maternal nutrition and/or fetal growth category, and only leptin mRNA was marginally positively related to fetal fat status.

Assessing prenatal fat status in analogous growth-restricted human fetuses *in utero* is technically challenging [50] and there is a paucity of information expressing any index of fat status on a fetal weight or birthweight specific basis. Yajnik [51] postulated that the higher relative adiposity observed in small and otherwise thin Indian babies (based on skinfold thickness) is

an adaptation directed at preserving growth of the predominantly lipid-containing fetal brain, with clear advantages for optimizing survival neonatally. The higher brain:liver weight ratio observed in the relatively fat and growth-restricted ovine fetuses here supports this hypothesis.

## Prenatal growth-restriction and postnatal lipid metabolism

The consequences of an altered prenatal growth trajectory, and the associated adaptations in adipose tissue development for lipid metabolism and body composition postnatally, were evaluated in contemporaneously treated offspring. Rather than reporting a snapshot at a single age, serial measures in the present study allowed a robust assessment across the life-course. Accordingly, relative to normally-grown controls, the growth-restricted offspring gestated by over-nourished dams had evidence of reduced insulin sensitivity in adipose tissue, namely higher fasting NEFA concentrations and attenuated insulin-induced inhibition of lipolysis during a glucose tolerance test (higher AUC NEFA), emerging while the lambs were rapidly growing and still suckling at 7 weeks of age, and, becoming highly pronounced by adolescent life. Although the magnitude of the differences between growth-restricted and normal offspring at each age varied slightly by gender, ultimately this altered lipid metabolism persisted to mid-adulthood in both sexes. At this stage, relative to normal females, the growth-restricted females were fatter (rate of fat accrual and body fat % from DEXA), had a greater relative PAT mass and higher expression of selected molecular markers of adipocyte function (discussed below). By contrast male offspring still had the potential to accumulate further body fat and although there was a trend for higher plasma leptin concentrations in growth-restricted males from adolescence forwards they were not markedly fatter than normal males by mid-adult life. Furthermore, we failed to observe any prenatal growth-related change in the expression of genes involved in adipocyte proliferation or function in adult males at study end. Nevertheless, given the evidence of insulin resistance at this stage (higher NEFA AUC and lower plasma adiponectin: leptin ratio), and our previously reported glucose intolerance from adolescence forwards in the same cohort (higher glucose AUC after glucose bolus, [35]), it is probable that these males would become obese in the longer term. Similar alterations in lipid metabolism during a glucose tolerance test have been reported at 2.5 years (but not 1.5years) in ovine offspring of both sexes following a short exposure to severe undernutrition during the first month of gestation [52]. The latter exposure had no effect on birth weight in either sex [53] and it is intriguing that models targeting such different periods of gestation, and with a contrasting influence on fetal growth, have a similar metabolic phenotype in the adult offspring. Nonetheless, in the human literature it is prenatal growth-restriction leading to low birthweight, as modelled here, that most consistently relates to an increased risk of insulin resistance, type 2 diabetes and greater adiposity at stages across the life-course, particularly when postnatal nutrient supply is abundant [54–62]: in adults these conditions are all typified by higher fasting and postprandial NEFA concentrations [63–65], with raised levels considered an early indicator of disease progression. In our animal model, plasma insulin concentrations (fasting and glucose-stimulated) were initially greater in restricted compared with normal birthweight lambs during the suckling and adolescent life stages, but there was no evidence of a primary β-cell deficiency in terms of insulin secretion by study-end in mid-adult life [35]. Thus, the higher NEFA concentrations in prenatally growth-restricted offspring are mainly reflecting reduced adipose tissue insulin sensitivity rather than any long-term change in insulin secretion. A similarly reduced capacity of insulin to suppress free fatty acids production during a rigorous hyperinsulinaemic-euglycaemic clamp carried out early in adult life has been measured in small-for-gestational-age humans (SGA, birthweight $<3^{rd}$ centile, [54]), and was likewise independent of any major impairment in insulin secretion. In the present study, greater NEFA concentrations (fasting

and AUC during the GTT) were evident at 7 weeks postnatal suggesting that altered lipid metabolism is an early presage of adverse metabolic health in the low birthweight animals. High peripheral lipid concentrations and other indicators of insulin resistance (measured directly and indirectly) are correspondingly reported in babies and young children defined as SGA at birth but the risk of exhibiting these metabolic disease indicators is largely confined to individuals who display rapid catch-up growth in early postnatal life [59]. Ninety-one percent of the prenatally growth-restricted animals studied here had faster relative growth compared with normally grown controls during the suckling phase and as a group they failed to completely catch-up by study end (modestly lower height and weight, [35]). Accordingly, both birthweight and the relative rate of early growth were strongly associated with lipid metabolism/insulin sensitivity during the glucose tolerance test close to study end.

The higher circulating NEFA concentrations in growth-restricted versus control female offspring at each age studied was matched by higher PAT expression of the lipolytic *HSL* gene at study end. Similarly, the greater body fat percentage and PAT mass in these growth-restricted females was reflected by higher gene expression of key adipokines, namely leptin and adiponectin. While the elevated *PPARγ* expression is arguably commensurate with continuing adipose tissue expansion in growth-restricted females, the unchanged expression of genes involved in fat storage, namely *G3PDH*, *LPL* and *FASN* suggests that maximum lipogenesis had already been achieved by mid-adult life. Indeed, the DEXA measurement indicates that two thirds of the growth-restricted females had exceeded fifty percent body fat by this stage. Others have reported an increase in adipose tissue *LPL* expression in similarly aged sheep exposed to early gestation undernutrition: in this case the difference was specific to males and independent of any observed change in body fatness [52]. Clearly measuring gene expression in a single fat depot and on a single occasion only provides a snapshot of the underlying molecular physiology. Moreover, our focus here was the adipose tissue and we are cognisant that expression may differ in other insulin-sensitive tissues such as skeletal muscle and liver.

### Fetal/offspring sex, adiposity and perirenal fat gene expression

To our knowledge this is the first ovine study sufficiently powered at the outset to enable the impact of fetal sex on adipose tissue phenotype and gene expression to be measured. Accordingly, relative to male fetuses, females had a higher body-weight-specific PAT mass with more multilocular adipocytes, and PAT leptin gene expression was greater irrespective of prenatal growth category. The latter confirms a prior (but underpowered) observation in the *undernourished model* [22]. This sexual dimorphism in ovine prenatal adipose tissue development and adipokine gene expression is perhaps unsurprising given that female sex in humans is associated with higher leptin concentrations in amniotic fluid at 16 weeks gestation and in cord blood at delivery [66,67], and with greater percentage body fat neonatally [68]. Leptin is likely to underlie the sex difference in adipocyte phenotype reported herein as leptin infusion for 4 days in late gestation sheep fetuses (of unspecified sex) increased the proportion of multilocular cells in PAT, while the amount of UCP1 protein also tended to be higher [69]. Notably, within the growth-restricted fetuses studied here, females had the highest *UCP1* gene expression suggestive of a temporal advance in adipose tissue development and potential thermogenic capacity compared with similarly-sized males. Indeed, the greater PAT expression of *IGF1* in males, a gene we have shown to be most highly expressed at an earlier stage of adipocyte proliferation and differentiation in mid-gestation [22], further aligns with a relative delay in adipose tissue development in this gender. These sex differences in fetal fat status may play a role in the lower Appearance, Pulse, Grimace, Activity and Respiration (APGAR) scores and poorer neonatal outcomes observed in male babies across settings [70–73]. The origin of the

dimorphism in prenatal adipose tissue development in both sheep and humans is likely to be differences in circulating sex steroid concentrations as the gonads of both species are steroidogenically-active from the point of sexual differentiation in early pregnancy [74,75]. Oestrogen receptors (ERα, ERβ) are present in the brown fat of human fetuses at mid- pregnancy, and *ERα* expression increases with gestational age over the range 15 to 23 weeks suggestive of a role for oestrogen in adipose tissue differentiation [76]. Similarly, the visceral fat of sheep fetuses express these receptors at mid (day 90, [77]) and later (day120, [78]) gestation and abundance is enhanced in females when fetal oestrogen concentrations are increased as a consequence of maternal testosterone treatment (model of polycystic ovary syndrome, [77,79]).

The effect of female sex on adiposity extends across the life-course. Regression analysis of serial DEXA data in contemporaneously exposed offspring in the current postnatal study revealed that females were approximately 70% fatter than males at birth, independent of prenatal growth. This wholly aligns with prior observations of greater internal fat mass, chemical carcass fat content, adipocyte size and expression of genes involved in lipogenesis and adipokine signalling in females compared with males necropsied at 11 weeks postnatal age [29,30], and supports the concept that females partition more nutrients into fat from the earliest stages of adipose tissue development. Given the large sex-difference in adiposity by the time the lambs are born, it is unsurprising that plasma leptin concentrations diverged soon after and were persistently higher in females, reflecting their greater body fat throughout most of the life-course. Maximum leptin concentrations were reached in females at an earlier stage of adult life than in males in line with reported sex differences in the rate of skeletal growth, fat deposition, bone density accrual and the attainment of mature body size [35, current study]. Thus, females reach a smaller adult size but are closer to maximum body fatness earlier than males. In contrast, even at two years of age and relative to normal females, normal birthweight males are continuing to deposit fat as indicated by their higher rate of fat accrual overall, albeit from a lower baseline, and greater body weight specific renal fat mass at necropsy. The robustly higher PAT expression of *PPARγ*, and genes involved in fat deposition (*LPL*, *FASN*), metabolism (*HSL*) and adipocyte signalling (leptin, adiponectin) in males at this age, irrespective of prenatal growth trajectory, corresponds with the premise of a still active and dynamic fat depot.

In summary, we have shown that growth-restricted fetuses have a relatively fat phenotype by late gestation and lipid metabolism is negatively perturbed across the life-course in both sexes when nutrients are abundant postnatally in contemporaneous animals. Sexual dimorphism in adiposity originates in fetal life and persists thereafter, and accordingly it is growth-restricted females that are overtly obese by mid-adult life. While the influence of offspring sex is more likely to be permanent the effect of prenatal growth-restriction on lipid metabolism may be amenable to postnatal dietary manipulation and requires to be tested.

## Supporting information

**S1 Table. Individual animal data for Experiment 1 (fetal study).**
(XLSX)

**S2 Table. Individual animal data for Experiment 2 (postnatal study).**
(XLSX)

## Author Contributions

**Conceptualization:** Jacqueline M. Wallace, Clare L. Adam.

**Formal analysis:** Jacqueline M. Wallace, John S. Milne.

**Funding acquisition:** Jacqueline M. Wallace, Clare L. Adam.

**Investigation:** Jacqueline M. Wallace, John S. Milne, Beth W. Aitken, Raymond P. Aitken, Clare L. Adam.

**Methodology:** Jacqueline M. Wallace, John S. Milne, Beth W. Aitken, Raymond P. Aitken, Clare L. Adam.

**Writing – original draft:** Jacqueline M. Wallace.

**Writing – review & editing:** Jacqueline M. Wallace, Clare L. Adam.

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
