## [Decision Letter · Decision Letter 0]

27 Dec 2019

PONE-D-19-32025

Prenatal growth-restriction and sex influence fetal adipose tissue phenotype and impact postnatal lipid metabolism and adiposity until adulthood.

PLOS ONE

Dear Dr Wallace,

Thank you for submitting your manuscript to PLOS ONE. After careful consideration, we feel that it has merit but does not fully meet PLOS ONE’s publication criteria as it currently stands. Therefore, we invite you to submit a revised version of the manuscript that addresses the points raised during the review process.

We would appreciate receiving your revised manuscript by Feb 10 2020 11:59PM. To enhance the reproducibility of your results, we recommend that if applicable you deposit your laboratory protocols in protocols.io, where a protocol can be assigned its own identifier (DOI) such that it can be cited independently in the future. For instructions see: http://journals.plos.org/plosone/s/submission-guidelines#loc-laboratory-protocols

We look forward to receiving your revised manuscript.

Kind regards,

Cristina Óvilo, Ph.D.

Academic Editor

PLOS ONE

Journal Requirements:

2. Please note that according to POS ONE policy, if materials, methods, and protocols are well established, authors may cite articles where those protocols are described in detail, but the submission should include sufficient information to be understood independent of these references (https://journals.plos.org/plosone/s/submission-guidelines#loc-materials-and-methods).Thus, we kindly request that you provide more information on the methods used (especially for embryo transfer) and detail the composition of the diet administered in your study.

3. According to our submissions guidelines, titles should be specific, descriptive, and concise; in this case, we suggest that your title is modified to clarify that all experiments were carried in vivo, and to specify the animal model used.

Reviewers' comments:

Reviewer's Responses to Questions

**Comments to the Author**

1. Is the manuscript technically sound, and do the data support the conclusions?

Reviewer #1: Yes

Reviewer #2: Yes

2. Has the statistical analysis been performed appropriately and rigorously? 

Reviewer #1: Yes

Reviewer #2: Yes

3. Have the authors made all data underlying the findings in their manuscript fully available?

Reviewer #1: No

Reviewer #2: Yes

4. Is the manuscript presented in an intelligible fashion and written in standard English?

Reviewer #1: Yes

Reviewer #2: Yes

5. Review Comments to the Author

Reviewer #1: This study provides a robust and highly controlled model of prenatal growth-restriction obtained by maternal overnutrition to decipher the effects on adipose tissue development and selected gene expression with consequences for lipid metabolism and body composition. Rather than reporting a snapshot at a single age, serial measures across the life-course are provided. Most factors are kept constant in the adolescent sheep model (one sire, transfer of single high-quality embryos harvested from donors with a known nutritional and reproductive background into young primiparous adolescent recipients of equivalent age, weight and adiposity at conception), allowing the effects of prenatal nutrient supply and also of sex to be demonstrated.

Minor comments:

- species should be indicated in the abstract and title (ewes)

- it’s a pity that fetus-derived DLK1 expression was not include in the list of genes studied by qPCR (e.g., Nat Genet. 2016 Dec;48(12):1473-1480. doi: 10.1038/ng.3699.)

- the authors claimed for data availability. However, in the main text, means and standard errors are provided, but individual data are not included in supplementary files or repository.

Reviewer #2: This paper contributes new findings in a field that is well-known for authors. Excellent discussion and results of fetal gene expression are very interesting. Authors propose future lines of work, including more tissues and time points. I encourage them to follow them. However, some parts could be improved to increase its readability simplifying the manuscript and focusing on main ideas.

The abstract has too much information in some sentences. I think focusing on key ideas in the abstract would improve its readability. Adding ‘ovine model’ would also be better.

Good and informative introduction. However, its last parts about authors’ models and a very long and detailed second goal remember Materials & Methods.

M&M is a detailed section, but weighty. Using supplementary documents could be useful to increase readability. I also think it would be better to have diets in supplementary data than in another paper.

I think using bold type in tables is a great idea. In Results, there are averages from Experiment 2 without SEM values.

Further details:

Some keywords are also in the title.

In table 2, the meaning of symbols (*,ɤ, ß) within individual groups is difficult to understand only reading the table.

In table 3, letter b is in all the groups in PPAR, and I think restricted males should be a. The restricted female group has the highest value for UCP1, are you checking individual values looking for outliers?

L687-8: Close brackets

L727: APGAR is not previously defined

6. PLOS authors have the option to publish the peer review history of their article (what does this mean?). If published, this will include your full peer review and any attached files.

Reviewer #1: No

Reviewer #2: No

---

## [Author Response · Author response to Decision Letter 0]

10 Jan 2020

All detailed in uploaded response to reviewers document

---

## [Editor Report · Decision Letter 1]

23 Jan 2020

Ovine prenatal growth-restriction and sex influence fetal adipose tissue phenotype and impact postnatal lipid metabolism and adiposity in vivo from birth until adulthood.

PONE-D-19-32025R1

Dear Dr. Wallace,

We are pleased to inform you that your manuscript has been judged scientifically suitable for publication and will be formally accepted for publication once it complies with all outstanding technical requirements.

With kind regards,

Cristina Óvilo, Ph.D.

Academic Editor

PLOS ONE
---

## [Editor Report · Acceptance letter]

3 Feb 2020

PONE-D-19-32025R1 

Ovine prenatal growth-restriction and sex influence fetal adipose tissue phenotype and impact postnatal lipid metabolism and adiposity in vivo from birth until adulthood. 

Dear Dr. Wallace:

I am pleased to inform you that your manuscript has been deemed suitable for publication in PLOS ONE. Congratulations! Your manuscript is now with our production department. 

With kind regards,

on behalf of

Dr. Cristina Óvilo 

Academic Editor

PLOS ONE